# Clinicopathologic Determinants of Lymph Node Count and Prognostic Significance of Metastatic Lymph Node Ratio in Colorectal Cancer

**DOI:** 10.3390/diagnostics15232962

**Published:** 2025-11-22

**Authors:** Fatma Yildirim, Murat Sezak, Osman Bozbiyik, Pinar Gursoy, Basak Doganavsargil

**Affiliations:** 1Department of Pathology, Memorial Health Group, Cankaya 06520, Ankara, Türkiye; 2Department of Pathology, Faculty of Medicine, Ege University, Bornova 35100, Izmir, Türkiye; drsezak@gmail.com (M.S.); bdoganavsargil@yahoo.com (B.D.); 3Department of General Surgery, Faculty of Medicine, Ege University, Bornova 35100, Izmir, Türkiye; bozbiyiko@gmail.com; 4Department of Medical Oncology, Faculty of Medicine, Ege University, Bornova 35100, Izmir, Türkiye; pinargursoy77@gmail.com

**Keywords:** colorectal cancer, lymph node count, metastatic lymph node ratio, prognosis, clinicopathologic factors, survival analysis

## Abstract

**Background/Objectives**: Accurate lymph node (LN) evaluation is crucial to predicting outcomes in colorectal cancer (CRC). Higher lymph node counts (LNCs) improve prognosis, whereas increased metastatic involvement worsens survival. This study aimed to identify factors associated with higher LNCs and evaluate the prognostic value of the metastatic lymph node ratio (MLNR). **Methods**: A retrospective analysis was performed on 989 CRC resections. Patients were stratified into four MLNR categories—MLNR0 (no metastasis), MLNR1 (<0.20), MLNR2 (0.20–0.50), and MLNR3 (>0.50)—and into two LNC groups—lower LNC (<12) and higher LNC (≥12). **Results**: The median LN count was 14 (range: 5–198). Lower LNCs occurred in 346 cases (35.0%), predominantly in the left colon. Higher LNCs were significantly associated with younger age (*p* < 0.001), larger tumor size (*p* < 0.001), higher pN stage (*p* < 0.001), right-sided location (*p* = 0.003), Crohn’s-like lymphocytic response (*p* = 0.006), and the absence of satellite nodules (*p* = 0.016). There were 86 pT4 and 178 pN2 tumors. Overall survival was 50.6%, with the 1-, 3-, and 5-year rates being 0.891, 0.721, and 0.612, respectively. Survival was higher in patients with higher LNCs (53.5% vs. 45.1%, *p* < 0.001). Survival rates by MLNR were 61.2% (MLNR0), 47.7% (MLNR1), 34.0% (MLNR2), and 26.4% (MLNR3). Mortality strongly correlated with MLNR (*p* < 0.001), and life expectancy decreased as MLNR increased (*p* < 0.01). **Conclusions**: MLNR provides superior prognostic information compared to pN status, even in patients with suboptimal lymph node retrieval (LNC < 12). As an independent survival predictor, MLNR may be integrated into staging systems and guide therapeutic strategies, highlighting its clinical utility in both standard and “gray zone” CRC cases.

## 1. Introduction

Colorectal cancer (CRC) remains one of the most significant global health challenges, currently ranking as the third most commonly diagnosed cancer and the second leading cause of cancer-related mortality worldwide [1]. Surgery is usually the primary treatment for curable colorectal cancers [2]. CRC surgery requires removal of the affected bowel as well as removal of the regional lymph nodes around the affected segment of bowel. Adequate lymphadenectomy is an important component of surgical treatment for non-metastatic CRC [3].

The stage at diagnosis is the most critical determinant in the management and prognosis of CRC patients. As defined by the eighth edition of the American Joint Committee on Cancer (AJCC) TNM staging system, the detection of lymph node metastasis advances the disease classification from Stage II to Stage III, indicating a more advanced and aggressive disease course [4]. Accurate staging is essential as it directly influences treatment decisions. Adjuvant chemotherapy is typically recommended for CRC patients with lymph node involvement (Stage III), as well as for those with Stage II disease who lack nodal metastasis but exhibit high-risk pathological features, including poor tumor differentiation, lymphovascular invasion, or perineural invasion [5]. Therefore, lymph node involvement determines disease stage, prognosis, and potential indication for adjuvant strategies. Thus, adequate lymph node resection and evaluation are required to accurately determine the stage of CRC. According to the AJCC guidelines, evaluation of at least 12 lymph nodes is recommended to meet the threshold requirement for adequate lymph node resection [4].

Nevertheless, the number of lymph nodes removed during CRC surgery may be insufficient in some patients, making patient prognosis assessment and further management more difficult. Recent studies have reported that CRC patients with a higher number of lymph nodes in the pathologic specimen have better overall survival (OS), especially in stage III disease [6,7].

The number of metastatic lymph nodes is known to be one of the most important factors in staging colon cancer and for predicting prognosis. At present, only the number of metastatic lymph nodes is incorporated into the TNM system [4]. This system, however, has been criticized for oversimplification. One of the most important limitations of the pN category within the TNM system is the effect of the total number of lymph nodes examined. When an insufficient number of lymph nodes are harvested, the number of metastatic lymph nodes may be underdetected, and this creates the potential for downstage migration. A ratio-based node staging system has been proposed in the literature to minimize this limitation. The metastatic lymph node ratio (MLNR) is defined as the proportion of involved lymph nodes to the total number of lymph nodes examined during surgical resection [8]. This ratio serves as a significant prognostic factor in colorectal cancer, providing crucial insights into the aggressiveness of the disease and informing postoperative management decisions [9]. A higher MLNR has been associated with poorer outcomes, including increased risk of recurrence and reduced overall survival rates [8,9].

In this study, we examined a large population-based cohort of patients undergoing colectomy for cancer to assess the clinicopathologic factors influencing the number of lymph nodes harvested and the prognostic impact of MLNR versus pN classification in the TNM staging system.

## 2. Materials and Methods

A total of 989 patients who underwent surgery for colorectal adenocarcinoma between 2002 and 2012, and whose resection specimens were examined at the Department of Pathology, Ege University Faculty of Medicine, were included in this study. Patients with rectal tumors who had received neoadjuvant chemotherapy were excluded. Clinicopathological and macroscopic data were retrieved from the hospital’s information system, while survival data were obtained from the Cancer Registry Center (KIDEM) of the Izmir Public Health Directorate. This study was conducted in accordance with the principles of the Declaration of Helsinki. Ethical approval was obtained from the Ethics Committee of Ege University, Faculty of Medicine (Decision No: 12-5.1/4; Date: 3 July 2012). Written informed consent was obtained from all participants prior to their inclusion in the study.

All patients were managed at a single tertiary center according to standardized institutional protocols. Surgical procedures included standardized lymphadenectomy, adjuvant chemotherapy was administered based on national guidelines, and follow-up protocols were consistently applied during the study period.

Histopathological evaluation was performed retrospectively on hematoxylin and eosin (H&E)-stained sections prepared from formalin-fixed, paraffin-embedded tissue blocks. Tumor invasion depth (pT) and lymph node metastasis status (pN) were recorded according to the 8th edition of the AJCC/UICC TNM classification [4]. Histological assessments were re-evaluated in accordance with the 2019 World Health Organization (WHO) criteria [10].

The following parameters were evaluated: histological tumor type, tumor grade, satellite tumor deposits, lymphovascular invasion (LVI), perineural invasion (PNI), presence of tumor-infiltrating lymphocytes (TILs), Crohn’s-like lymphoid reaction, dirty necrosis, tumor budding, and tumor border configuration. Tumors were also assessed for mucinous, signet ring cell, medullary, and poorly differentiated components.

### Statistical Analysis

The primary outcome variables of the study were the lymph node count (LNC) and, as a subheading, the metastatic lymph node ratio. Patients were divided into quartiles according to the distribution of MLNR values. While the 50th percentile (median) value was 0, the 75th percentile (3rd quartile) value was 20% and the 90th percentile value was 50%. In the next stage, the cases were divided into four categories—those without metastases, MLNR < 0.20, MLNR 0.20–0.50, and MLNR > 0.50—and were harmonized with the clinical findings. These MLNR thresholds were largely consistent with those reported in previous studies evaluating MLNR in colorectal cancer, ensuring comparability across different cohorts. Quartile-based MLNR cut-offs were chosen to allow for reproducible stratification and comparability with prior studies; alternative statistical approaches, such as maximally selected rank statistics or externally validated thresholds, may be explored in future work. The secondary endpoint of the study was OS, which was calculated as the time from the patient’s operation until death from any cause.

Clinicopathologic factors that may be effective at differentiating groups with LN counts as the lower LNC group (<12 LNs) and the higher LNC group (≥12 LNs) were analyzed by univariate logistic regression analysis. Factors with *p* < 0.10 as a result of univariate statistical analysis were included in the multivariate logistic regression model as candidate factors. Then, the factors with the most significant determinants of LNC were identified by considering the forward stepwise elimination (Forward LR) procedure. Odds ratios, 95% confidence intervals, and Wald statistics were calculated for all variables.

One-way analysis of variance (one-way ANOVA) was used to analyze the significance of the differences between the groups in terms of continuous numerical variables for which the assumptions of parametric test statistics were met. If the results of one-way analysis of variance were found to be significant, the post hoc Tukey HSD test was used to determine the condition(s) causing the difference. The significance of the differences between the groups in terms of continuous numerical variables for which the assumptions of parametric test statistics were not met was evaluated by the Kruskal–Wallis test.

Kaplan–Meier survival analysis using the log-rank test was used to evaluate whether there was a statistically significant change in OS according to the LNC and MLNR. OS rates in each subgroup; 1-year, 3-year, and 5-year survival rates; mean life expectancy; and 95% confidence intervals were also calculated.

The univariate effects of all possible variables thought to affect OS were examined with Cox’s proportional hazards regression models. Then, the factor(s) with the most significant effect on OS were investigated with multivariate Cox’s proportional hazards models using the forward stepwise elimination (Forward LR) method. All variables with *p* < 0.10 after univariate statistical analysis were included in the regression model as candidate risk factors. In addition, hazard ratios (HRs), 95% confidence intervals, and Wald statistics were calculated for each variable.

Data were analyzed using IBM SPSS Statistics 25.0 (IBM Corporation, Armonk, NY, USA). Results were considered statistically significant for *p* < 0.05.

## 3. Results

In the present study, data from 989 patients aged between 22 and 94 years were evaluated. The mean age was 62.5 ± 12.7 (years), 593 patients (60.0%) were male, and 396 (40.0%) were female. The mean tumor diameter was 5.25 ± 2.37 (cm).

The median number of lymph nodes removed was 14, with a minimum of 5 and a maximum of 198 lymph nodes per case. In 346 cases (35.0%) LNC was less than 12 and, in the remaining 643 cases (65.0%), more than 12 LNs were detected. The median follow-up period was 71 months. During this follow-up interval, 489 (49.4%) of the patients died and the OS rate was 50.6%.

Table 1 demonstrates a comparison of the clinicopathologic features of the cases with their lymph node counts.

The LNC decreased statistically significantly with advancing age (OR = 0.977, 95% CI: 0.966–0.987, *p* < 0.001). Tumor localization on the right led to a higher LNC (OR = 2.016, 95% CI: 1.430–2.841, *p* < 0.001), while tumor localization on the left led to a lower LNC (OR = 0.587, 95% CI: 0.446–0.772, *p* < 0.001). As the tumor size increased, LNC also increased statistically significantly (OR = 1.196, 95% CI: 1.118–1.280, *p* < 0.001). Poorly differentiated adenocarcinomas were statistically more likely to have a higher LNC than well-differentiated adenocarcinomas (OR = 2.167, 95% CI: 1.117–4.204, *p* = 0.022). The LNC was also statistically significantly higher in pN2 tumors compared to pN0 tumors (OR = 1.934, 95% CI: 1.298–2.882, *p* < 0.001). Among the histologic features, only the presence of a Crohn’s-like reaction statistically significantly increased the probability of having a higher LNC (OR = 1.464, 95% CI: 1.113–1.926, *p* = 0.006). There were no significant correlations between a higher LNC and gender, tumor type, T stage, LVI, PNI, satellite tumor deposit, presence of TIL, mucinous tumor component, signet ring cell component, medullary tumor component, presence of tumor budding, and tumor border, respectively (*p* > 0.05).

All variables with *p* < 0.10 after univariate statistical analysis were included in the regression model as candidate risk factors. As a result of prospective elimination, the most predictive factors for a higher LNC were tumor diameter, age, pN classification, localization, Crohn’s-like reaction, and satellite tumor deposit, respectively.

In Table 2, the clinicopathologic features that most influenced the number of lymph nodes being ≥12 were determined by multivariate logistic regression analysis.

Independent of other factors, a larger tumor size (OR = 1.176, 95% CI: 1.097–1.262, *p* < 0.001), younger patient (OR = 1.236, 95% CI: 1.106–1.384, *p* < 0.001), pN2 stage (OR = 2.349, 95% CI: 1.491–3.701, *p* < 0.001), right colon localization (OR = 1.723, 95% CI: 1.201–2. 471, *p* = 0.003), Crohn’s-like lymphocytic reaction (OR = 1.501, 95% CI: 1.126–2.000, *p* = 0.006), and the absence of satellite tumor deposit (OR = 1.733, 95% CI: 1.107–2.710, *p* = 0.016) statistically significantly increased the likelihood of a higher LNC.

Table 3 shows the Kaplan–Meier results examining the effect of the LNC and metastatic lymph node ratio on OS.

The OS rate was 50.6%, while the 1-, 3-, and 5-year cumulative survival rates were 0.891, 0.721, and 0.612, respectively (Figure 1). The median life expectancy among all cases was 103.6 months (95% CI: 98.6–108.5).

The OS rate was 45.1% in lower LNC cases and 53.5% in higher LNC cases. The 1-, 3-, and 5-year cumulative survival rates were 0.859, 0.667, and 0.547 in lower LNC cases and 0.908, 0.749, and 0.645 in higher LNC cases, respectively (Figure 2). The median life expectancy was 93.2 months (95% CI: 84.9–101.6) in lower LNC cases and 108.7 months (95% CI: 102.6–114.8) in higher LNC cases. According to the results, the prognosis was statistically significantly worse in lower LNC cases (log-rank = 10.559 and *p* < 0.001).

The overall survival rate of patients with no MLN was 61.2%, whereas the corresponding rates were 47.7%, 34.0%, and 26.4% in the groups with MLNR < 0.20, 0.20–0.50, and >0.50, respectively. The 1-, 3-, and 5-year cumulative survival rates for no MLN were 0.921, 0.835, and 0.743, respectively; meanwhile, the corresponding rates were 0.911, 0.725, and 0.616 in patients with MLNR <0.20, respectively. The 1-, 3-, and 5-year cumulative survival rates were 0.830, 0.543, and 0.396, respectively, for patients with an MLNR between 0.20 and 0.50, and 0.777, 0.346, and 0.205, respectively, for patients with an MLNR > 0.50 (Figure 3). There was a statistically significant change in mortality rates depending on MLNR (log-rank = 135.141, *p* < 0.001), and life expectancy decreased from the group with no MLNR to the group with MLNR >0.50 (*p* < 0.01).

Table 4 shows the results of univariate Cox’s proportional hazards regression analysis of the effect of clinicopathologic factors on overall survival.

As a result of univariate statistical analysis, all variables with *p* < 0.10 were included in the regression model as candidate risk factors. As a result of prospective elimination, the factors that had the most significant effect on overall survival were metastatic lymph node ratio, age, LVI, PNI, mucinous tumor component, number of lymph nodes, tumor budding, pT stage, and Crohn’s-like reaction, respectively. In Table 5, the most significant factors on overall survival were determined by multivariate Cox’s proportional hazards regression analysis.

In the two-step multivariate analyses, both pN stage and MLNR were included in the first step as predictors of prognosis. In the second step, the prognostic significance of pN stage was lost, whereas MLNR retained its predictive value, highlighting its superior role in survival estimation.

## 4. Discussion

In this study, we found that the independent factors predicting the likelihood of an LNC ≥ 12 in multivariate analyses were younger age, larger tumor diameter, increased pN category, right colon localization, Crohn’s-like reaction, and absence of satellite tumor deposits.

In our study cohort, the mortality rate was significantly increased in lower LNC tumors compared to higher LNC tumors, and also in MLNR group 3 (>0.50) tumors compared to MLNR group 0 tumors.

A two-step multivariate analysis was conducted to compare the prognostic value of MLNR categories with that of traditional pN categories. In the first step, pN categories emerged as independent prognostic factors. However, in the second step, the prognostic significance of pN categories was lost and was replaced by MLNR categories. This pattern was consistent in both higher LNC and lower LNC groups. These findings suggest that MLNR categories offer improved prognostic value compared to pN categories, regardless of whether the minimum recommended number of lymph nodes was retrieved.

The AJCC TNM classification remains the most reliable prognostic indicator for patients with colorectal cancer. In the seventh edition of the TNM classification, the AJCC introduced a more detailed subdivision of the pN category for colon cancer [11]. This included N1a (1 positive lymph node), N1b (2–3 positive lymph nodes), N1c (no positive lymph nodes but the presence of tumor deposits in the subserosa, mesentery, or nonperitonealized pericolic or perirectal tissues), N2a (4–6 positive lymph nodes), and N2b (7 or more positive lymph nodes) [11]. Furthermore, this classification framework has been retained without modification in the eighth edition of the TNM system. Importantly, neither the LNCs examined nor the MLNR were incorporated into the TNM staging system. MLNR represents the likelihood of positive nodes among the examined lymph nodes and is relatively independent of the total number of nodes collected. This characteristic minimizes the limitations related to the number of lymph nodes required for accurate pN category staging.

It has been reported in the literature that MLNR is more effective at predicting prognostic results in CRC patients. Yang et al. developed a novel staging system (TNRM) that integrates the LNR with the AJCC TNM classification to improve prognostic accuracy in CRC patients with fewer than 12 retrieved lymph nodes [12]. Their study, based on a SEER training cohort and an independent validation cohort, demonstrated that the TNRM system offers superior survival stratification and successfully addresses the survival paradox seen in the AJCC system, suggesting enhanced prognostic value over conventional AJCC staging [12]. Derwinger et al., Berger et al., Lee et al., and Wang et al. reported that MLNR was a highly significant prognostic factor in CRC by using quartiles to divide LNRs into four groups [13,14,15,16,17]. Chen et al. conducted an analysis of 36,712 colon cancer patients from the National Cancer Institute’s SEER database [18]. Consistent with our results, they found that MLNR provided the most accurate prognostic information when at least 12 lymph nodes were examined [18]. Furthermore, their multivariate analysis indicated that MLNR was a superior prognostic indicator compared to the traditional pN classification system.

However, while these studies highlight the potential of MLNR as a viable alternative to pN categories, they also reveal significant gaps in the current understanding of optimal thresholds for MLNR that could guide clinical practice. Many studies have calculated MLNR using the quartile method, which we also applied in our study. However, Tong et al. used running log-rank statistics to determine two optimal cutoff values for MLNR and proposed an rN category [8]. These categories demonstrated significant prognostic differences in both colon and rectal cancers, regardless of whether lymph node retrieval was sufficient or insufficient, and were identified as independent prognostic factors for overall survival. Their findings indicate that the rN categories are effective at predicting prognosis in colorectal cancer.

In addition, studies conducted in different populations have also stated that MLNR is an adaptable approach to cancer staging that takes into account regional differences in tumor biology, thus emphasizing its importance in different demographic groups. Building on this, MLNR may be particularly useful in cases within the “gray zone,” where the current TNM staging system is less comprehensive, particularly for patients with fewer than 12 examined lymph nodes. MLNR can be considered in addition to the pN stage. A higher MLNR is likely to be associated with a worse prognosis, which may inform clinicians to select more intensive adjuvant therapy. Thus, MLNR provides additional granularity for risk stratification and may guide personalized treatment decisions in patients who are otherwise challenging to classify.

The number of sampled and histologically analyzed LNs plays an important role, not only as an independent prognostic marker for therapeutic decisions, but also as a marker for adequate staging, quality of surgery, and pathologic analysis [3].

Accurate evaluation of lymph node yield in CRC is fundamentally influenced by a multitude of underlying clinicopathological factors that intersect with tumor biology and patient characteristics. An intricate understanding of these factors enables clinicians to optimize surgical and pathological protocols, directly impacting staging accuracy and therapeutic decision-making.

Previous research has demonstrated that the total LNC decreases with increasing patient age at the time of diagnosis [19]. This could be due to differences in immune response or tumor biology. It has been hypothesized that age-related immune senescence may contribute to the observed decline in LNC among older patients [20]. Tumors in younger patients often exhibit more aggressive behavior, including higher rates of lymphovascular invasion [21]. This can stimulate the surrounding lymphatic tissue, leading to an increased number of reactive lymph nodes. Lymph node hyperplasia may also enhance LN identification by pathologists, as larger nodes are more readily detected during specimen evaluation. In younger patients, a more robust immune response may promote increased LN hyperplasia, thereby facilitating greater lymph node retrieval [22]. In addition, differences between surgical procedures performed on older and younger patients may also affect the number of lymph nodes. Younger patients may undergo more extensive surgical procedures due to their ability to tolerate aggressive treatments [23]. This can result in a higher lymph node yield.

As tumor size increases, the likelihood of a higher lymph node yield tends to rise. Larger tumors often stimulate lymphangiogenesis, the formation of new lymphatic vessels, as part of their invasive growth [22]. This biological response increases the likelihood of retrieving more lymph nodes during surgical resection. Larger masses typically require more extensive surgical procedures, which can also lead to a higher lymph node yield.

Most of studies suggest that right-sided tumors may result in more lymph nodes being harvested compared to left-sided tumors [22,23]. The difference in lymph node harvest between right-sided and left-sided colorectal tumors is influenced by several factors, such as embryological and anatomical differences, tumor histological features, surgical techniques, and pathological examinations. Right-sided tumors occur in the proximal colon, which originates from the midgut during embryological development [22,24]. This region has a richer lymphatic network compared to the distal colon, which develops from the hindgut. Consequently, right-sided tumors are associated with higher lymph node yields. Right-sided tumors often exhibit distinct molecular characteristics, such as microsatellite instability (MSI) and BRAF mutations [25]. Microsatellite instability-high (MSI-H) tumors, which are more common in younger patients, are associated with a strong immune response and higher lymph node counts [25]. These features can lead to increased lymphangiogenesis, contributing to higher lymph node counts. The surgical approach for right-sided tumors typically involves more extensive lymphadenectomy due to the anatomical layout and the need to ensure complete removal of affected lymphatic tissue [24]. Therefore, pathologists may retrieve more lymph nodes from specimens of right-sided tumors due to their larger size and the presence of more reactive lymph nodes.

The pN category in the TNM staging system reflects the extent of lymph node metastasis [4]. Patients with a higher pN category (e.g., pN2) generally have a worse prognosis due to the greater extent of lymphatic spread. In our study, we found that the number of lymph nodes increased as the pN stage increased. However, in our literature review, we did not find any studies that had found a correlation between the pN category and the lymph node count. Tumors with higher pN stages often exhibit aggressive biological behavior such as increased lymphovascular invasion and higher proliferative activity [26]. These factors contribute to the spread of cancer cells to regional lymph nodes. When the immune system detects tumor antigens, it can trigger an immune response that activates lymphocytes within regional lymph nodes. This activation can cause lymph nodes to enlarge and become more prominent, making it easier to identify and retrieve them during surgical resection.

Crohn’s-like reactions (CLRs) can contribute to an increased lymph node count in colorectal cancer. CLR is characterized by peritumoral lymphoid aggregates, which are clusters of immune cells at the advancing edge of the tumor [25]. These aggregates represent an adaptive immune response to the tumor and can lead to lymphoid hyperplasia, increasing the number of lymph nodes retrieved during surgical resection.

Satellite tumor deposits were first described as discrete clusters of cancer cells in peritumoral soft tissue, separate from the primary tumor and not within identifiable lymph nodes [27]. However, in the eighth edition of its staging system, the AJCC redefined tumor deposits as discrete tumor nodules located in the lymphatic drainage area of the primary carcinoma, lacking identifiable lymph node tissue and discernible vascular or neural structures [28]. In our study, we found an inverse correlation between satellite tumor deposits and LNC. To our knowledge, this is one of the first studies to demonstrate an inverse association between tumor deposits and lymph node count.

In a molecular study examining the relationship between tumor deposits and lymph node metastasis, transcriptome analyses from tumor deposits revealed increased cell motility, matrix remodeling, and epithelial–mesenchymal transition (EMT)-related gene expression [29]. In this study, transcriptomic and proteomic analyses point toward tumor deposits having a more invasive, mesenchymal, and fibrotic nature [29]. EMT is a reversible cellular process in which steady-state epithelial cancer cells transform into motile and invasive mesenchymal-like cells, ultimately leading to circulating tumor cells [30]. It has been reported in the literature that EMT is an important process critical for immune resistance, but it is also a powerful driver for the activation of an immunosuppressive network within the tumor microenvironment [31]. Therefore, immunosuppression in the tumor microenvironment prevents stimulation of the lymph nodes around the tumor and leads to a lower LNC.

## 5. Conclusions

In conclusion, this study found that the most important clinicopathological factors affecting lymph node count were a lower age, larger tumor diameter, higher pN category, right colon localization, Crohn’s-like reaction, and absence of satellite tumor nodules. To the best of our knowledge, this is the first study to demonstrate a significant association between lymph node count and key histopathological parameters such as a Crohn’s-like reaction and the presence of satellite tumor deposits.

The present study highlights the prognostic relevance of the metastatic lymph node ratio (MLNR) and its potential to complement traditional pN staging, particularly in cases with fewer than 12 examined lymph nodes. The findings also emphasize the inverse relationship between lymph node count and satellite tumor deposits, and the possible interplay between Crohn’s-like lymphocytic responses and metastatic behavior. Collectively, these observations support the use of MLNR as an additional, easily applicable prognostic tool that may refine staging accuracy and guide therapeutic decisions in colorectal cancer.

Nevertheless, several limitations should be acknowledged. The retrospective, single-center nature of this study may limit the generalizability of the findings. Disease-specific survival data were not available for all patients, and overall survival was used as the primary endpoint. In addition, molecular data, such as microsatellite instability and KRAS/BRAF mutation status, were not routinely assessed during the study period and therefore could not be included in the analysis. Despite these limitations, the large cohort size, consistent pathological assessment, and comprehensive statistical approach strengthen the validity and clinical relevance of our findings.

## Figures and Tables

**Figure 1 diagnostics-15-02962-f001:**
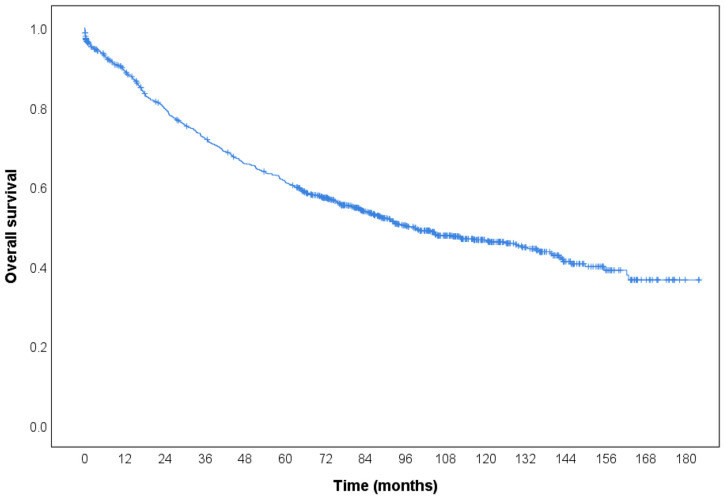
Kaplan–Meier curve demonstrating overall survival in 989 patients. The overall survival rate was 50.6%, with 1-, 3-, and 5-year cumulative survival rates of 89.1%, 72.1%, and 61.2%, respectively. The median overall survival was 103.6 months (95% CI: 98.6–108.5).

**Figure 2 diagnostics-15-02962-f002:**
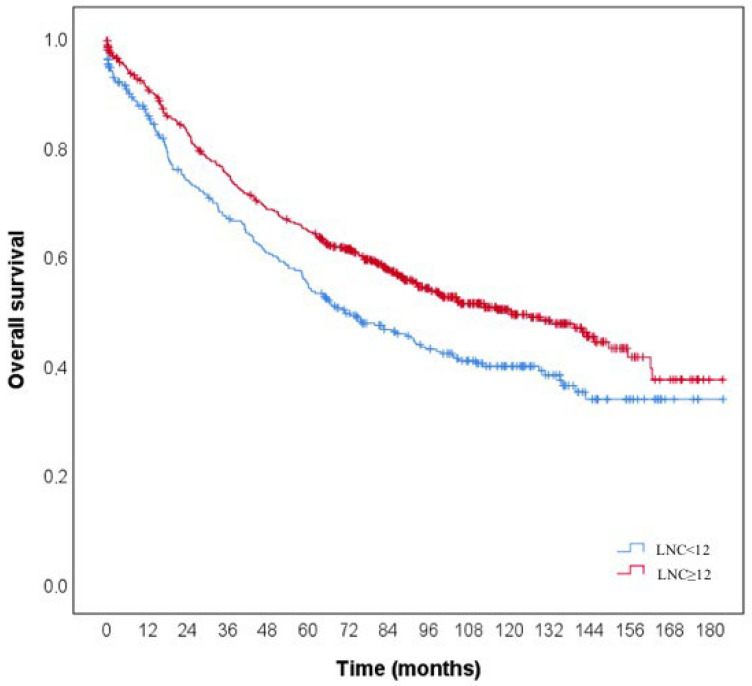
Kaplan–Meier curves comparing overall survival by lymph node count (LNC). Median survival was 93.2 months in lower LNC cases and 108.7 months in higher LNC cases. Prognosis was significantly worse in lower LNC cases (log-rank = 10.559, *p* < 0.001).

**Figure 3 diagnostics-15-02962-f003:**
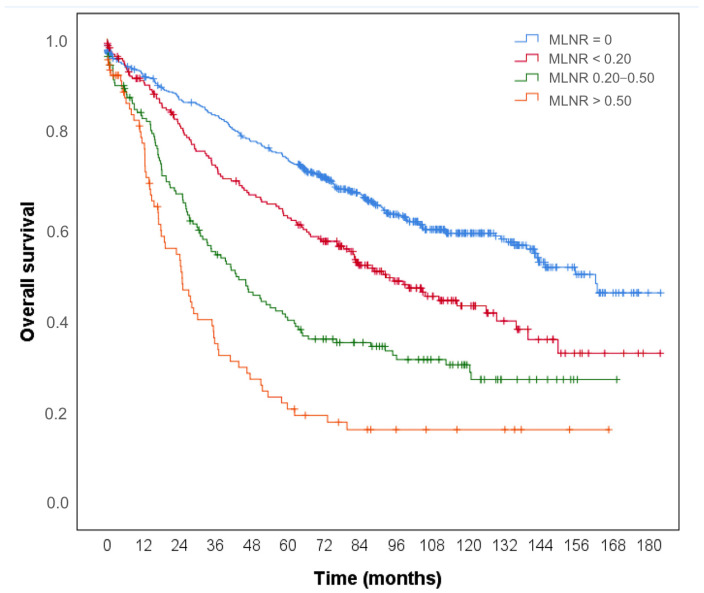
Kaplan–Meier curves showing overall survival stratified by metastatic lymph node ratio (MLNR). Survival decreased stepwise from patients with no MLN to those with MLNR > 0.50 (log-rank = 135.141, *p* < 0.001).

**Table 1 diagnostics-15-02962-t001:** Comparison of the clinicopathological features of the cases according to lymph node count.

	Total (*n* = 989)	LNC < 12 (*n* = 346)	LNC ≥ 12 (*n* = 643)	OR (95% CI)	*p*-Value
Age	62.5 ± 12.7	64.9 ± 11.4	61.2 ± 13.2	0.977 (0.966–0.987)	<0.001
Gender					
Male	593 (60.0%)	208 (60.1%)	385 (59.9%)	1.000	-
Female	396 (40.0%)	138 (39.9%)	258 (40.1%)	1.010 (0.774–1.319)	0.941
Tumor localization					
Rectum	446 (45.1%)	153 (44.2%)	293 (45.6%)	1.056 (0.812–1.373)	0.685
Right colon	221 (22.3%)	52 (15.0%)	169 (26.3%)	2.016 (1.430–2.841)	<0.001
Left colon	326 (33.0%)	141 (40.8%)	185 (28.8%)	0.587 (0.446–0.772)	<0.001
Tumor size	5.25 ± 2.37	4.71 ± 2.46	5.54 ± 2.27	1.196 (1.118–1.280)	<0.001
Tumor type					
Adenocarcinoma	885 (89.5%)	312 (90.2%)	573 (89.1%)	1.000	-
Mucinous	98 (9.9%)	33 (9.5%)	65 (10.1%)	1.073 (0.690–1.667)	0.756
Signet ring cell	6 (0.6%)	1 (0.3%)	5 (0.8%)	2.723 (0.317–23.406)	0.362
Tumor differentiation					
Well	93 (10.5%)	39 (12.5%)	54 (9.4%)	1.000	-
Moderate	716 (80.9%)	254 (81.4%)	462 (80.6%)	1.314 (0.947–2.038)	0.224
Poor	76 (8.6%)	19 (6.1%)	57 (10.0%)	2.167 (1.117–4.204)	0.022
pT stage					
pT_1_	13 (1.3%)	6 (1.7%)	7 (1.1%)	1.000	-
pT_2_	96 (9.7%)	46 (13.3%)	50 (7.8%)	0.932 (0.292–2.977)	0.905
pT_3_	794 (80.3%)	268 (77.5%)	526 (81.8%)	1.682 (0.560–5.055)	0.354
pT_4_	86 (8.7%)	26 (7.5%)	60 (9.3%)	1.978 (0.606–6.460)	0.259
pN stage					
pN0	476 (48.1%)	171 (49.4%)	305 (47.4%)	1.000	-
pN1	296 (29.9%)	118 (34.1%)	178 (27.7%)	0.946 (0.627–1.140)	0.272
pN1c	39 (3.9%)	17 (4.9%)	22 (3.4%)	0.726 (0.375–1.404)	0.341
pN2	178 (18.0%)	40 (11.6%)	138 (21.5%)	1.934 (1.298–2.882)	<0.001
LVI	248 (25.1%)	86 (24.9%)	162 (25.2%)	1.018 (0.753–1.377)	0.907
PNI	233 (23.6%)	87 (25.1%)	146 (22.7%)	0.875 (0.645–1.186)	0.389
Satellite tumor deposit	172 (17.4%)	70 (20.2%)	102 (15.9%)	0.743 (0.531–1.041)	0.085
Presence of TIL	103 (10.4%)	28 (8.1%)	75 (11.7%)	1.500 (0.951–2.364)	0.081
Crohn’s-like reaction	380 (38.4%)	113 (32.7%)	267 (41.5%)	1.464 (1.113–1.926)	0.006
Mucinous tumor component					
Absent	648 (65.5%)	227 (65.6%)	421 (65.5%)	1.000	-
<50%	236 (23.9%)	85 (24.6%)	151 (23.5%)	0.958 (0.702–1.307)	0.786
≥50%	105 (10.6%)	34 (9.8%)	71 (11.0%)	1.126 (0.726–1.747)	0.597
Signet ring cell component	39 (3.9%)	11 (3.2%)	28 (4.4%)	1.387 (0.682–2.820)	0.367
Medullary tumor component	49 (5.0%)	11 (3.2%)	38 (5.9%)	1.913 (0.965–3.792)	0.063
Tumor budding	234 (23.7%)	84 (24.3%)	150 (23.3%)	0.949 (0.699–1.289)	0.738
Tumor border					
Expansive	71 (7.2%)	28 (8.1%)	43 (6.7%)	1.000	-
Infiltrative	918 (92.8)	318 (91.9%)	600 (93.3%)	1.229 (0.749–2.016)	0.415

OR: odds ratio, CI: confidence interval, LNC: lymph node count, LVI: lymphovascular invasion, PNI: perineural invasion, TIL: tumor-infiltrating lymphocyte.

**Table 2 diagnostics-15-02962-t002:** The most influential clinicopathologic features on lymph node count ≥12 by multivariate logistic regression analysis results.

	Odds Ratio	95% Confidence Interval		Wald	*p*-Value
		Lower Bound	Upper Bound		
Age *	1.236	1.106	1.384	14.408	<0.001
Right colon localization	1.723	1.201	2.471	8.718	0.003
Tumor size	1.176	1.097	1.262	20.787	<0.001
N1	1.004	0.721	1.398	0.001	0.980
N1c	1.372	0.607	3.100	0.577	0.447
N2	2.349	1.491	3.701	13.564	<0.001
Absence of satellite tumor deposit	1.733	1.107	2.710	5.792	0.016
Crohn’s-like reaction	1.501	1.126	2.000	7.684	0.006

* The effect of each 10-year decrease in age on the LNC.

**Table 3 diagnostics-15-02962-t003:** The effects of the lymph node count (LNC) and metastatic lymph node ratio (MLNR) on overall survival.

	Cases	Number of Deaths	Overall Survival Rate	Cumulative Survival Rate			Estimated Life Expectancy (Month) *	Log-Rank	*p*-Value
				1 year	3 years	5 years			
LNC								10.559	<0.001
<12	346	190	45.1%	0.859	0.667	0.547	93.2 (84.9–101.6)		
≥12	643	299	53.5%	0.908	0.749	0.645	108.7 (102.6–114.8)		
MLNR								135.141	<0.001
MLNR 0	518	201	61.2%	0.921	0.835	0.743	122.8 (116.3–129.3) ^A^		
MLNR 1 (<0.20)	218	114	47.7%	0.911	0.725	0.616	100.0 (89.8–110.2) ^B^		
MLNR 2 (0.20–0.50)	162	107	34.0%	0.830	0.543	0.396	71.2 (60.6–81.7) ^C^		
MLNR 3 (>0.50)	91	67	26.4%	0.777	0.346	0.205	47.1 (34.9–59.3) ^D^		
LNC < 12 cases								83.534	<0.001
MLN 0	190	81	57.4%	0.903	0.823	0.721	118.3 (107.6–129.0) ^A^		
MLNR 1 (<0.20)	45	31	31.1%	0.909	0.602	0.445	68.6 (51.3–85.8) ^B^		
MLNR 2 (0.20–0.50)	69	46	33.3%	0.823	0.526	0.378	68.4 (52.3–84.5) ^B^		
MLNR 3 (>0.50)	42	32	23.8%	0.650	0.187	0.062	27.8 (14.7–41.0) ^C^		
LNC ≥ 12 cases								64.552	<0.001
MLN 0	328	120	63.4%	0.932	0.843	0.756	123.7 (115.9–131.5) ^A^		
MLNR 1 (<0.20)	173	83	52.0%	0.911	0.754	0.656	107.8 (96.2–119.4) ^B^		
MLNR 2 (0.20–0.50)	93	61	34.4%	0.835	0.556	0.408	68.9 (56.4–81.4) ^C^		
MLNR 3 (>0.50)	49	35	28.6%	0.874	0.460	0.307	58.1 (42.0–74.3) ^C^		
MLN Absent								1.648	0.199
LNC < 12	190	81	57.4%	0.903	0.823	0.721	118.3 (107.6–129.0)		
LNC ≥ 12	328	120	63.4%	0.932	0.843	0.756	123.7 (115.9–131.5)		
MLNR < 0.20								9.839	0.002
LNC < 12	45	31	31.1%	0.909	0.602	0.445	68.6 (51.3–85.8)		
LNC ≥ 12	173	83	52.0%	0.911	0.754	0.656	107.8 (96.2–119.4)		
MLNR 0.20–0.50								0.181	0.670
LNC < 12	69	46	33.3%	0.823	0.526	0.378	68.4 (52.3–84.5)		
LNC ≥ 12	93	61	34.4%	0.835	0.556	0.408	68.9 (56.4–81.4)		
MLNR > 0.50								10.042	0.002
LNC < 12	42	32	23.8%	0.650	0.187	0.062	27.8 (14.7–41.0)		
LNC ≥ 12	49	35	28.6%	0.874	0.460	0.307	58.1 (42.0–74.3)		
Total	989	489	50.6%	0.891	0.721	0.612	103.6 (98.6–108.5)	-	-

LNC: lymph node count, MLN: metastatic lymph node, MLNR: metastatic lymph node ratio. * Descriptive statistics are shown as mean (95% confidence interval). Differences between groups with different capital letters were statistically significant (*p* < 0.01).

**Table 4 diagnostics-15-02962-t004:** Results of univariate Cox’s proportional hazards regression analysis of clinicopathologic characteristics affecting overall survival.

	HR (95% CI)	Wald	*p*-Value
Age	1.021 (1.013–1.029)	29.302	<0.001
Male gender factor	1.123 (0.936–1.348)	1.563	0.211
Right colon localization	1.149 (0.929–1.422)	1.634	0.201
Tumor size	1.010 (0.973–1.048)	0.256	0.613
Mucinous tumor type	1.411 (1.075–1.853)	6.144	0.013
Signet ring cell tumor type	1.867 (0.697–4.998)	1.543	0.214
Moderate differentiation	1.485 (1.039–2.123)	4.714	0.030
Poor differentiation	2.098 (1.330–3.311)	10.136	<0.001
pT2	1.843 (0.437–7.765)	0.694	0.405
pT3	4.339 (1.081–17.412)	4.287	0.038
pT4	8.087 (1.972–33.164)	8.427	0.004
Lymph node count < 12	1.351 (1.126–1.620)	10.476	<0.001
MLNR < 0.20	1.548 (1.230–1.949)	13.871	<0.001
MLNR 0.20–0.50	2.510 (1.983–3.178)	58.57	<0.001
MLNR > 0. 50	4.126 (3.117–5.462)	98.065	<0.001
pN1	1.931 (1.562–2.386)	36.999	<0.001
pN1c	1.826 (1.161–2.872)	6.802	0.009
pN2	3.078 (2.442–3.880)	90.528	<0.001
LVI	2.201 (1.821–2.662)	66.303	<0.001
PNI	2.123 (1.752–2.573)	59.044	<0.001
Satellite tumor deposit	2.068 (1.676–2.552)	45.888	<0.001
Absence of TILs	1.787 (1.267–2.521)	10.945	<0.001
Crohn’s-like reaction	0.765 (0.635–0.922)	7.916	0.005
Medullary tumor component	0.978 (0.637–1.500)	0.011	0.918
Tumor budding	1.529 (1.254–1.863)	17.684	<0.001
Infiltrative tumor border	1.613 (1.094–2.379)	5.828	0.016

HR: hazard ratio, CI: confidence interval, CI: confidence interval, MLNR: metastatic lymph node ratio, LVI: lymphovascular invasion, PNI: perineural invasion, TIL: tumor-infiltrating lymphocyte.

**Table 5 diagnostics-15-02962-t005:** Clinicopathologic features most predictive of overall survival—multivariate Cox’s proportional hazards regression analysis results.

	HR (95% CI)	Wald	*p*-Value
Age	1.026 (1.018–1.034)	42.486	<0.001
pT_2_	1.650 (0.391–6.967)	0.464	0.496
pT_3_	3.304 (0.821–13.298)	2.829	0.093
pT_4_	5.410 (1.312–22.311)	5.454	0.020
Lymph node count < 12	1.326 (1.096–1.604)	8.448	0.004
MLNR < 0.20	1.331 (1.049–1.689)	5.532	0.019
MLNR 0.20–0.50	1.919 (1.488–2.475)	25.204	<0.001
MLNR > 0.50	2.972 (2.193–4.026)	49.392	<0.001
LVI	1.525 (1.238–1.880)	15.670	<0.001
PNI	1.479 (1.201–1.821)	13.557	<0.001
Crohn’s-like reaction	0.801 (0.662–0.969)	5.238	0.022
Mucinous tumor component < 50%	0.981 (0.788–1.220)	0.030	0.862
Mucinous tumor component ≥ 50%	1.660 (1.258–2.192)	12.794	<0.001
Tumor budding	1.324 (1.078–1.626)	7.140	0.008

HR: hazard ratio, CI: confidence interval, LVI: lymphovascular invasion, PNI: perineural invasion.

## Data Availability

The data presented in this study are available on request from the corresponding author. The data are not publicly available due to privacy or ethical restrictions.

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
