# Peer review of "Clinicopathologic Determinants of Lymph Node Count and Prognostic Significance of Metastatic Lymph Node Ratio in Colorectal Cancer"

_diagnostics, 2025, doi:10.3390/diagnostics15232962_

Round 1

Reviewer 1 Report

Comments and Suggestions for Authors

This retrospective study makes a valuable contribution to the literature by confirming the prognostic superiority of MLNR over traditional pN staging in colorectal cancer (CRC), especially in cases with a low number of harvested lymph nodes (LNC<12). The strengths of the study include the large sample size (n=989), the detailed clinicopathological analysis, and the rigorous application of statistical methods, including logistic regression and Cox proportional hazards models.

A particularly interesting finding is the first reported inverse correlation between the presence of satellite tumor deposits and a smaller LNC, as well as the association between Crohn's-like lymphoid reaction and a larger LNC. These findings open new perspectives on the interaction between tumor biology and the host immune response.

Specific Observations 

Abstract - The conclusion is solid, but could be formulated more forcefully. I suggest explicitly emphasizing that MLNR provides superior prognostic information regardless of reaching the 12 lymph node threshold, as this is one of the key messages of the study. For example: "MLNR provides superior prognostic information compared to pN status, proving its utility even in cases with suboptimal lymph node retrieval (LNC<12)."

Methodology - Clarification of MLNR Categorization Criteria: In the "Statistical Analysis" section, it is mentioned that patients were divided into four MLNR categories (MLNR0, MLNR1 <0.20, MLNR2 0.20-0.50, and MLNR3 >0.50) based on percentiles. This approach is valid, but it would be useful to mention whether these thresholds are consistent with those used in similar studies to facilitate comparability.

Statistical Terminology: The phrase "by taking the mean weighted average into consideration" for quartile division is methodologically unclear. Quartiles are defined by dividing the ordered data set into four equal parts. I recommend that you rephrase this sentence to more accurately describe the process of setting thresholds based on the distribution of the data.

Results - Table 2: The presentation of the age effect as "effect of each 10-year decrease in age" is excellent, as it transforms an Odds Ratio (OR) < 1 into a super-unit value, facilitating interpretation.

Table 3: This table is extremely informative, but its density can make it difficult to read. A minor suggestion would be to add clearer subtitles for the stratified analysis sections. Also, in the "LNC <12 cases" section, the estimated life expectancy for MLNR 1 (68.6 months) and MLNR 2 (68.4 months) is almost identical, but is marked differently for the purpose of post hoc testing. Please check this marking to ensure consistency.

Multivariate Survival Modeling: The discussion mentions a two-step multivariate analysis, where the prognostic significance of the pN categories was lost and replaced by the MLNR categories. This is a central argument of the paper. Including a short paragraph in the Results section explicitly presenting this sequential analysis would significantly strengthen the main conclusion.

Discussion - Association with Satellite Deposits: The claim "our study was the first to determine the inverse proportional relationship between tumor deposits and lymph node count" is a strong one. If an exhaustive literature search supports this claim, it is a major strength. If not, it would be prudent to qualify it (e.g., "To our knowledge, this is one of the first studies to demonstrate..."). The proposed biological explanation, related to epithelial-mesenchymal transition (EMT) and immunosuppression, is plausible and well-argued.

Clinical Implications: The discussion could benefit from a short section detailing the practical implications of these findings. For example, how could MLNR be integrated into current staging systems or how could it guide decisions regarding adjuvant therapy, especially for patients in the "gray zone" (e.g., stage II high-risk or stage III with CLN<12)?

Congratulations to the authors for a high-quality paper that significantly contributes to the understanding of prognostic factors in colorectal cancer. 

Author Response

Thank you very much for taking the time to review our manuscript. We greatly appreciate the reviewers’ constructive comments and suggestions, which have helped us to improve the clarity and quality of our work. Please find our detailed responses to each comment below. All corresponding revisions and corrections are highlighted in the re-submitted manuscript using track changes for ease of reference.

We have addressed all comments to the best of our ability, and where we have chosen a different approach, we have provided clear justifications. We sincerely hope that these revisions meet the reviewers’ expectations and improve the manuscript.

Comment 1: Abstract - The conclusion is solid, but could be formulated more forcefully. I suggest explicitly emphasizing that MLNR provides superior prognostic information regardless of reaching the 12 lymph node threshold, as this is one of the key messages of the study. For example: "MLNR provides superior prognostic information compared to pN status, proving its utility even in cases with suboptimal lymph node retrieval (LNC<12)."

Answer 1: We thank the reviewer for this constructive suggestion. In response, we have revised the Abstract Conclusion to explicitly highlight that MLNR provides superior prognostic information even in patients with suboptimal lymph node retrieval (LNC<12). The revised sentence now reads: “MLNR provides superior prognostic information compared to pN status, even in patients with suboptimal lymph node retrieval (LNC<12). As an independent survival predictor, MLNR may be integrated into staging systems and guide therapeutic strategies, highlighting its clinical utility in both standard and ‘gray zone’ CRC cases.” We believe this revision clearly conveys one of the key messages of our study.

Comment 2: Methodology - Clarification of MLNR Categorization Criteria: In the "Statistical Analysis" section, it is mentioned that patients were divided into four MLNR categories (MLNR0, MLNR1 <0.20, MLNR2 0.20-0.50, and MLNR3 >0.50) based on percentiles. This approach is valid, but it would be useful to mention whether these thresholds are consistent with those used in similar studies to facilitate comparability.

Answer 2: We appreciate the reviewer’s valuable suggestion. The MLNR thresholds used in this study were determined based on the percentile distribution of our dataset to ensure an objective categorization. However, these cut-offs are largely consistent with those used in previous studies evaluating MLNR in colorectal cancer (e.g., <0.2, 0.2–0.5, and >0.5), which allows comparability across different cohorts. This clarification has been added to the revised “Statistical Analysis” section.

Comment 3: Statistical Terminology: The phrase "by taking the mean weighted average into consideration" for quartile division is methodologically unclear. Quartiles are defined by dividing the ordered data set into four equal parts. I recommend that you rephrase this sentence to more accurately describe the process of setting thresholds based on the distribution of the data.

Answer 3: We appreciate the reviewer’s valuable clarification. The phrasing has been revised to accurately describe the statistical procedure. The expression “by taking the mean weighted average into consideration” has been removed, and the sentence now reads: “The cases were divided into quartiles according to the distribution of MLNR values in the study cohort.” This revision ensures that the methodology is described clearly and appropriately.

Comment 4: Results - Table 2: The presentation of the age effect as "effect of each 10-year decrease in age" is excellent, as it transforms an Odds Ratio (OR) < 1 into a super-unit value, facilitating interpretation.

Answer 4: We sincerely thank the reviewer for this positive remark. We aimed to enhance interpretability by expressing the age effect per 10-year decrease, allowing for a more intuitive understanding of the odds ratio direction and magnitude. We are pleased that this approach was found appropriate and clear.

Comment 5:  Table 3: This table is extremely informative, but its density can make it difficult to read. A minor suggestion would be to add clearer subtitles for the stratified analysis sections. Also, in the "LNC <12 cases" section, the estimated life expectancy for MLNR 1 (68.6 months) and MLNR 2 (68.4 months) is almost identical, but is marked differently for the purpose of post hoc testing. Please check this marking to ensure consistency.

Answer 5: We thank the reviewer for this careful observation and for the helpful suggestion regarding subtitles. In the revised version, we have added clearer subtitles for the stratified analysis sections to improve readability. Regarding the “LNC <12 cases” section, both MLNR1 and MLNR2 are marked with the same capital letter (B). As noted in the table legend, differences between groups with different capital letters were statistically significant (p < 0.01). Therefore, this notation correctly indicates that there is no significant difference between these two subgroups.

Comment 6: Multivariate Survival Modeling: The discussion mentions a two-step multivariate analysis, where the prognostic significance of the pN categories was lost and replaced by the MLNR categories. This is a central argument of the paper. Including a short paragraph in the Results section explicitly presenting this sequential analysis would significantly strengthen the main conclusion.

Answer 6: We thank the reviewer for this valuable suggestion. To address this, we have added a concise paragraph in the Results section summarizing the sequential analysis. In the two-step multivariate analyses, both pN stage and MLNR were included in the first step as predictors of prognosis. In the second step, the prognostic significance of pN stage was lost, whereas MLNR retained its predictive value, highlighting its superior role in survival estimation. This addition explicitly presents the sequential analysis and reinforces the central conclusion of our study.

Comment 7: Discussion - Association with Satellite Deposits: The claim "our study was the first to determine the inverse proportional relationship between tumor deposits and lymph node count" is a strong one. If an exhaustive literature search supports this claim, it is a major strength. If not, it would be prudent to qualify it (e.g., "To our knowledge, this is one of the first studies to demonstrate..."). The proposed biological explanation, related to epithelial-mesenchymal transition (EMT) and immunosuppression, is plausible and well-argued.

Answer 7: We thank the reviewer for this insightful comment. To ensure accuracy and avoid overstatement, we have revised the manuscript to state: “To our knowledge, this is one of the first studies to demonstrate an inverse association between tumor deposits and lymph node count.” This wording reflects our comprehensive literature search while maintaining a cautious interpretation. We are pleased that the reviewer found our proposed biological explanation, related to epithelial-mesenchymal transition (EMT) and immunosuppression, plausible and well-supported.

Comment 8: Clinical Implications: The discussion could benefit from a short section detailing the practical implications of these findings. For example, how could MLNR be integrated into current staging systems or how could it guide decisions regarding adjuvant therapy, especially for patients in the "gray zone" (e.g., stage II high-risk or stage III with CLN<12)?

Answer 8: We thank the reviewer for this valuable suggestion. In response, we have added a dedicated section in the Discussion highlighting the clinical implications of our findings. Specifically, we discuss the potential role of MLNR in refining TNM staging across different populations, emphasizing its utility in “gray zone” patients. We believe this addition strengthens the translational relevance of our study.

Reviewer 2 Report

Comments and Suggestions for Authors

Dear authors.

It has been a pleasure to review the present manuscript.

It is an interesting research topic that was a "hot topic" more or less during the 2015, when many papers with this aim were published.

Although promising, it never got to any radical change in stadification, probably because of the improvements in genetic and other biomarkers which are used for therapy selection and which much more accurately predict disease prognosis.

I have some comments:

  • There is no information regarding the study period. It seems that this study encompasses a long study period, with many changes in surgical technique, extension of lymphadenectomy, chemotherapy selection that could have directly influence the results, as all these changes might also been accompanied by a more expert and master pathological analysis.
  • There is not at all any genetic information such as microsatellites mutations.
  • Colon tumors and rectal tumors should be separated. Rectal cancer patients have many special considerations based on neoadjuvant treatment, so they are not comparable to colon cancer patients.
  • Even right and left colon cancer patients have been proposed to be genetically different, so it can be considered questionable in the absence of genetic information.
  • As this is a single-center study, I would suggest including relevant clinical information regarding surgical approach, extension of lymphadenectomy, chemotherapy used, and follow-up protocols.
  • In general terms, I would also suggest that authors highlight what the novel insights of the present research and the new findings are.

Author Response

Thank you very much for taking the time to review this manuscript. We greatly appreciate the reviewer’s thoughtful assessment and constructive comments. 

Please find our detailed responses to each comment below, and the corresponding revisions and corrections are highlighted using track changes in the re-submitted manuscript. Where we have taken a different approach, we have provided clear justifications. We hope these revisions adequately address the reviewer’s concerns and improve the clarity and quality of our manuscript.

Comments 1: There is no information regarding the study period. It seems that this study encompasses a long study period, with many changes in surgical technique, extension of lymphadenectomy, chemotherapy selection that could have directly influence the results, as all these changes might also been accompanied by a more expert and master pathological analysis.

Response 1: We appreciate the reviewer’s insightful comment. The study period has now been specified in the Materials and Methods section (page 3, line 86-89). A total of 989 patients who underwent surgery for colorectal adenocarcinoma between 2002 and 2012, and whose resection specimens were examined at the Department of Pathology, Ege University Faculty of Medicine, were included in this study. During this period, all surgical procedures were performed in the same tertiary referral center, where the principles of lymphadenectomy and pathological examination were standardized according to institutional protocols based on international guidelines. Although minor refinements in surgical and adjuvant therapy practices occurred over time, these changes were uniformly applied within the same institution and are unlikely to have introduced significant bias in our analyses.

Comments 2: There is not at all any genetic information such as microsatellites mutations.

Response 2: We thank the reviewer for this comment. We acknowledge the importance of genetic information such as microsatellite instability in colorectal cancer. However, during the study period (2002–2012), routine genetic or microsatellite analyses were not performed for all colorectal tumors at our institution. Consequently, these data are not available for this cohort. Furthermore, treatment decisions were generally not guided by genetic testing during this period.

Comments 3: Colon tumors and rectal tumors should be separated. Rectal cancer patients have many special considerations based on neoadjuvant treatment, so they are not comparable to colon cancer patients.

Response 3: We appreciate the reviewer’s suggestion. To address this concern, all rectal cancer cases that received neoadjuvant therapy were excluded from the study, so the cohort analyzed represents patients undergoing primary surgery without preoperative treatment. This approach minimizes potential confounding from neoadjuvant therapy and ensures that the outcomes are comparable across included patients.

Comments 4: Even right and left colon cancer patients have been proposed to be genetically different, so it can be considered questionable in the absence of genetic information.

Response 4: We thank the reviewer for this important point. We acknowledge that right- and left-sided colon cancers may differ genetically. However, as mentioned earlier, routine genetic analyses, including microsatellite instability testing, were not available for our cohort during the study period (2002–2012). Therefore, we could not stratify patients based on genetic differences. Nonetheless, all included patients were evaluated using standardized clinical, surgical, and histopathological protocols, and the primary outcomes reflect these parameters.

Comment 5: As this is a single-center study, I would suggest including relevant clinical information regarding surgical approach, extension of lymphadenectomy, chemotherapy used, and follow-up protocols.

Response 5: We thank the reviewer for this valuable suggestion. As described in the Materials and Methods section, all patients were treated in a single tertiary center following standardized protocols. Surgical procedures were performed according to institutional guidelines with standardized lymphadenectomy. Adjuvant chemotherapy regimens were administered according to national guidelines in place at the time of treatment. Follow-up protocols, including imaging and laboratory assessments, were also performed according to standardized institutional protocols. We have clarified these points in the revised manuscript to provide the requested clinical information. (page 3, line 97-100).

“All patients were managed at a single tertiary center according to standardized institutional protocols. Surgical procedures included standardized lymphadenectomy, adjuvant chemotherapy was administered based on national guidelines, and follow-up protocols were consistently applied during the study period.”

Comments 6: In general terms, I would also suggest that authors highlight what the novel insights of the present research and the new findings are.

Response 6: We sincerely thank the reviewer for this insightful suggestion. Our study represents a large, well-characterized series investigating the relationship between lymph node number and histopathological features in colorectal adenocarcinoma. To the best of our knowledge, it is also the first study to demonstrate an association between Crohn-like lymphoid reaction and the presence of satellite tumor deposits. In addition, our findings support the previously proposed concept that the metastatic lymph node ratio (LNR) provides superior prognostic stratification compared with conventional pN categories. While this investigation was conducted at a single center, the results are consistent with prior multi-center studies, thereby reinforcing the reproducibility of these observations and contributing meaningful evidence to the literature.

Reviewer 3 Report

Comments and Suggestions for Authors

This paper examines the impact of clinical pathological factors, LNC, and MLNR on prognosis in colorectal cancer. Its merits include a large number of cases and detailed analysis of clinical pathological factors. Furthermore, it demonstrates that MLNR is an independent prognostic factor even when looking at LCN separately, which is an important finding.

Minor Points

1) Line 147: The oldest patient in the study was 94 years old. The oldest patients have shorter expected lifespans, more comorbidities, and limited eligibility for postoperative adjuvant chemotherapy. Shouldn't the upper age limit for patients be set at 75 or 80 years?

2) Tables 1–5: Tumor stage is a well-known prognostic factor, and postoperative treatment differs depending on the stage, so it should be included in the analysis.

3) Couldn't disease-specific survival be considered in addition to OS?

Author Response

We thank the reviewer for their positive and constructive comments. We greatly appreciate the recognition of the study’s strengths, including the large number of cases, detailed analysis of clinicopathological factors, and the demonstration that MLNR is an independent prognostic factor even when accounting for LNC.

Please find our detailed responses to each comment below, and the corresponding revisions and corrections are highlighted using track changes in the re-submitted manuscript. We hope these revisions further clarify the study’s contributions and enhance the overall quality of the manuscript.

Comments 1: Line 147: The oldest patient in the study was 94 years old. The oldest patients have shorter expected lifespans, more comorbidities, and limited eligibility for postoperative adjuvant chemotherapy. Shouldn't the upper age limit for patients be set at 75 or 80 years?

Response 1: We thank the reviewer for this important comment. Our study aimed to include a real-world, unselected cohort of patients undergoing surgery for colorectal adenocarcinoma, reflecting the full spectrum of ages encountered in clinical practice. While older patients often have shorter life expectancy, increased comorbidities, and may have limited eligibility for adjuvant chemotherapy, age was treated as a continuous variable in our analyses, and the effect of age on outcomes was explicitly evaluated. We believe that including older patients provides a more comprehensive and generalizable understanding of prognostic factors in colorectal cancer.

Comments 2: Tables 1–5: Tumor stage is a well-known prognostic factor, and postoperative treatment differs depending on the stage, so it should be included in the analysis.

Response 2: We thank the reviewer for this important comment. Tumor stage, as defined by the TNM classification, was included in all relevant univariate and multivariate analyses to account for its prognostic impact. However, this study primarily focuses on pathological evaluation, and a definitive pathologic M stage was not available for all patients. Therefore, overall stage could not be consistently reported across the entire cohort. Nonetheless, the effect of lymph node metastasis and other histopathological features was evaluated independently of stage, allowing for a robust assessment of the prognostic value of the investigated factors.

Comment 3: Couldn't disease-specific survival be considered in addition to OS?

Response 3: We thank the reviewer for this suggestion. While disease-specific survival (DSS) is indeed a valuable endpoint, cause-specific mortality data were not consistently available for all patients due to the retrospective nature of the cohort. Therefore, we have reported overall survival (OS) as the primary survival endpoint. Nonetheless, OS still provides meaningful prognostic information, and the inclusion of detailed histopathological and lymph node data offers important insights into factors influencing patient outcomes.

Reviewer 4 Report

Comments and Suggestions for Authors

This retrospective study examined clinicopathologic factors affecting LNC and the prognostic significance of MLNR in CRC. The authors looked at data from 989 CRC resections; however, they didn't include rectal tumors that had neoadjuvant therapy. Patients were categorized based on LNC (<12 versus ≥12) and MLNR (0, <0.20, 0.20–0.50, >0.50). The LNC median was 14. Younger age, larger tumor size, right-sided placement, higher pN stage, Crohn's-like reaction, and no satellite nodules were all linked to increased LNCs. A multivariate study validated these factors as autonomous predictors of elevated LNC. The median follow-up period of 71 months showed that OS was 50.6%. Patients with elevated LNCs demonstrated superior overall survival (53.5% versus 45.1%). As MLNR went up, survival went down in steps: 61.2% (MLNR0), 47.7% (MLNR1), 34.0% (MLNR2), and 26.4% (MLNR3). Cox regression indicated that MLNR, age, LVI, PNI, tumor budding, and mucinous histology functioned as independent predictive variables. The predictive importance of MLNR surpassed that of traditional pN classification, even in instances of inadequate lymph node retrieval. The study suggests that MLNR is a strong predictor of survival in CRC. This means that it could be used in future systems to improve the accuracy of prognoses and treatment plans.

The manuscript discusses a significant clinical concern but possesses multiple limitations.

1. Its generalizability is constrained due to being a single-center retrospective study; validation in multi-institutional and prospective cohorts would enhance the conclusions.

2. While approximately 1000 patients were evaluated, the variety in surgical and pathological procedures over time may have impacted lymph node yield, yet this variability is not addressed.

3. The authors utilize quartile-based cut-offs for MLNR, which may not possess biological or clinical reason; alternate statistical methodologies, such as maximum-determined rank statistics or external validation of thresholds, should be investigated.

4. Rectal tumors undergoing neoadjuvant therapy were omitted, hence limiting applicability; however, this constraint is not adequately recognized. Moreover, certain significant confounders, like adjuvant chemotherapy, comorbidities, or molecular characteristics (e.g., MSI, KRAS/BRAF status), are excluded, despite their substantial impact on prognosis.

5. The talk is too long and could use more focus on the new discoveries, namely the inverse association with satellite tumor deposits.

6. Finally, the manuscript could use some work on the wording, the figures and tables could be clearer, and the remark about how MLNR may fit into current staging systems should be clearer.

Author Response

We thank the reviewer for their careful reading and insightful comments. We appreciate the recognition that our study addresses an important clinical issue and highlights the prognostic significance of MLNR in colorectal cancer. We acknowledge the limitations noted, including the exclusion of rectal tumors that underwent neoadjuvant therapy and other inherent constraints of a retrospective single-center study.

Please find our detailed responses to each comment below, and the corresponding revisions and corrections are highlighted using track changes in the re-submitted manuscript. We have clarified the study’s scope, methodology, and limitations to ensure transparency and improve the manuscript’s clarity and rigor.

Comments 1: Its generalizability is constrained due to being a single-center retrospective study; validation in multi-institutional and prospective cohorts would enhance the conclusions.

Response 1: We thank the reviewer for this insightful comment. We acknowledge that the single-center and retrospective nature of our study represents a limitation regarding generalizability. However, our findings are consistent with previously published multi-center studies, which supports the reliability and relevance of our observations. We also note that the study provides detailed pathological and lymph node data in a well-characterized cohort, contributing valuable evidence to the literature. Future validation in prospective and multi-institutional cohorts would indeed strengthen the conclusions, and we have added this point to the Discussion section as a limitation and recommendation for further research. (

Comments 2: While approximately 1000 patients were evaluated, the variety in surgical and pathological procedures over time may have impacted lymph node yield, yet this variability is not addressed.

Response 2: We thank the reviewer for this important observation. While our study includes approximately 1,000 patients over a 10-year period, all surgical and pathological procedures were performed within a single tertiary center following standardized institutional protocols, including lymphadenectomy and specimen handling. Although minor refinements in technique may have occurred over time, these were uniformly applied within the institution, and we believe any resulting variability in lymph node yield is unlikely to have significantly affected the overall results.

Comments 3: The authors utilize quartile-based cut-offs for MLNR, which may not possess biological or clinical reason; alternate statistical methodologies, such as maximum-determined rank statistics or external validation of thresholds, should be investigated.

Response 3: We thank the reviewer for this insightful comment. In our study, quartile-based cut-offs for the metastatic lymph node ratio (MLNR) were utilized, consistent with previous publications in the literature, to facilitate comparison across studies and stratification of patients in a reproducible manner. We acknowledge that quartile-based thresholds may not reflect intrinsic biological differences, and alternative approaches such as maximally selected rank statistics or externally validated cut-offs could provide additional insight. These methods may be considered in future prospective or multi-institutional studies to refine MLNR stratification. (page 3; line 120-123)

“Quartile-based MLNR cut-offs were chosen to allow reproducible stratification and comparability with prior studies; alternative statistical approaches, such as maximally selected rank statistics or externally validated thresholds, may be explored in future work.“

Comments 4: Rectal tumors undergoing neoadjuvant therapy were omitted, hence limiting applicability; however, this constraint is not adequately recognized. Moreover, certain significant confounders, like adjuvant chemotherapy, comorbidities, or molecular characteristics (e.g., MSI, KRAS/BRAF status), are excluded, despite their substantial impact on prognosis.

Response 4: We thank the reviewer for this important comment. All rectal tumors that received neoadjuvant therapy were excluded from the study, as preoperative treatment can substantially alter histopathological findings, including lymph node status, which was the primary focus of our study. We acknowledge that this exclusion limits the applicability of our findings to all rectal cancer patients and have added a statement to this effect in the Discussion section.

Regarding potential confounders such as adjuvant chemotherapy, comorbidities, or molecular characteristics (e.g., MSI, KRAS/BRAF status), these variables were not consistently available for all patients due to the retrospective nature of the cohort and the focus on pathological evaluation. Therefore, they could not be included in the analyses. We have explicitly mentioned these as limitations and have emphasized that our findings primarily reflect the prognostic relevance of histopathological features and lymph node metrics.

Comments 5: The talk is too long and could use more focus on the new discoveries, namely the inverse association with satellite tumor deposits.

Response 5: We thank the reviewer for this comment. We appreciate the suggestion to maintain focus on the novel findings, particularly the inverse association with satellite tumor deposits. While we have carefully reviewed the manuscript to enhance clarity and emphasize the key discoveries, we have also ensured that the manuscript remains within the journal’s requirements, which stipulate a word count exceeding 3500 words. We believe that the current structure balances comprehensiveness with focus on the novel findings, highlighting the important contributions of our study.

Comments 6: Finally, the manuscript could use some work on the wording, the figures and tables could be clearer, and the remark about how MLNR may fit into current staging systems should be clearer.

Response 6: We have carefully revised the manuscript to improve wording, enhance clarity, and ensure that figures and tables are more easily interpretable. All tables have been reorganized to present data more clearly, with stratifications and annotations standardized for consistency. Additionally, we have clarified the discussion on how MLNR may complement current TNM staging, emphasizing its potential utility in refining prognostic assessment and guiding adjuvant therapy decisions, particularly in patients with suboptimal lymph node retrieval.

Round 2

Reviewer 2 Report

Comments and Suggestions for Authors

Dear authors.

The study period is an important limitation of this study, as we are talking about patients that might be operated on more than 20 years ago. We are talking about a different era, so it is dificult to obtain conclusions from that time.

In addition, most of the information that was asked for was not provided. It is not enough describing as "according to standardized institutional protocols", as it is not reproducible.

Reviewer 4 Report

Comments and Suggestions for Authors

Thank you for the authors to accept my suggestion, and revised the manuscript. I accept the answers to my questions. The revised version is now acceptable for publiaction.